# Statin adherence is lower in primary than secondary prevention: A national follow-up study of new users

**Finn Sigglekow**[1]ᵒ, **Simon Horsburgh**[1,2]ᵒ*, **Lianne Parkin**[1,2]ᵒ

**1** Department of Preventive and Social Medicine, Otago Medical School—Dunedin Campus, University of Otago, Dunedin, New Zealand, **2** Pharmacoepidemiology Research Network, University of Otago, Dunedin, New Zealand

ᵒ These authors contributed equally to this work.
* simon.horsburgh@otago.ac.nz

**Data Availability Statement:** The data underlying the results presented in the study are available from the following sources: 1) The New Zealand Ministry of Health (https://www.health.govt.nz/nz-health-statistics/access-and-use/how-access-

## Abstract

### Background

Maintaining adherence to statins reduces the risk of an initial cardiovascular disease (CVD) event in high-risk individuals (primary prevention) and additional CVD events following the first event (secondary prevention). The effectiveness of statin therapy is limited by the level of adherence maintained by the patient. We undertook a nationwide study to compare adherence and discontinuation in primary and secondary prevention patients.

### Methods

Dispensing data from New Zealand community pharmacies were used to identify patients who received their first statin dispensing between 2006 and 2011. The Medication Possession Ratio (MPR) and proportion who discontinued statin medication was calculated for the year following first statin dispensing for patients with a minimum of two dispensings. Adherence was defined as an MPR $\geq$ 0.8. Previous CVD was identified using hospital discharge records. Multivariable logistic regression was used to control for demographic and statin characteristics.

### Results

Between 2006 and 2011 289,666 new statin users were identified with 238,855 (82.5%) receiving the statin for primary prevention compared to 50,811 (17.5%) who received it for secondary prevention. The secondary prevention group was 1.55 (95% CI 1.51–1.59) times as likely to be adherent and 0.67 (95% CI 0.65–0.69) times as likely to discontinue statin treatment than the primary prevention group. An early gap in statin coverage increased the odds of discontinuing statin treatment.

### Conclusion

Adherence to statin medication is higher in secondary prevention than primary prevention. Within each group, a range of demographic and treatment factors further influences adherence.

data): The Pharmaceutical Collection (PHARMS), The National Minimum Dataset (Hospital Events) (NMDS), The National Health Index collection (NHI), The Mortality Collection (MORT). 2) The Medical Council of New Zealand (https://www.mcnz.org.nz/about-us/contact-us/): The Medical Practitioners' Registration database. Data can be accessed via request from the sources listed above. The authors did not have any special access to the data used in this study.

**Funding:** This research was funded by a Dunedin School of Medicine Summer Scholarship (FS) and a University of Otago Research Grant (SH and LP). The funders had no role in the study design in the collection, analysis and interpretation of data, in the writing of the report; or in the decision to submit the article for publication.

**Competing interests:** The authors have declared that no competing interests exist.

## Introduction

Cardiovascular disease (CVD) is one of the leading causes of morbidity and mortality in New Zealand [1]. HMG-CoA reductase inhibitors (statins) are effective at reducing lipid levels and preventing CVD [2–5]. Successive versions of New Zealand primary care guidelines have recommended statins for primary prevention in patients with no history of CVD disease if their 5-year CVD risk is above 15% using risk prediction equations based on the Framingham Heart Study [6], and for secondary prevention following angina, myocardial infarction, ischaemic stroke, or transient ischaemic attack [7–10]. Statins have become a mainstay of the pharmacological prevention of CVD in New Zealand; in 2015 514,277 people (14.6% of the New Zealand population aged 18 or over) were dispensed a statin and this number has continued to rise [11, 12].

The effectiveness of statin therapy is limited by a patient's level of adherence (the degree to which a patient conforms to a prescribed course of medication [13]). Research has demonstrated greater lipid control with higher statin adherence levels [14]. A protective relationship between statin adherence and CVD outcomes has also been found in both primary and secondary prevention populations in a number of observational studies [15–19]. In addition to the health outcome benefits accruing from the effectiveness of statin therapy, high levels of adherence lead to reduced future healthcare costs through decreased hospitalisations [20–22]. Collectively, improving statin adherence leads to better patient and health system outcomes.

The research undertaken in New Zealand to date has focused on statin adherence following an acute coronary event (i.e. secondary prevention) [23–25], and no studies internationally have compared adherence in primary and secondary prevention using the full population of statin users within a country. We undertook a large nationwide study of new users of statins to (i) compare adherence and discontinuation among patients prescribed statins for primary versus secondary prevention, (ii) explore adherence and discontinuation within primary and secondary prevention groups according to patient characteristics, and (iii) ascertain, within primary and secondary prevention groups, whether a delay in obtaining a second supply of a statin during the first year of statin therapy is associated with discontinuation.

## Materials and methods

### Data sources

The study used national demographic, pharmaceutical dispensing, and hospital discharge data supplied by the New Zealand Ministry of Health. Statin dispensing data came from the Pharmaceutical Collection database (PHARMS) which contains records of all claims by community-based pharmacists for the dispensing of prescription medications which are publicly funded [26]. Five statins were approved for use in New Zealand during the study period: atorvastatin, pravastatin, rosuvastatin, simvastatin, and a combination simvastatin and ezetimibe product. Rosuvastatin was not publicly funded so was not included in the PHARMS dispensing data. In 2010 the funding criteria for atorvastatin changed allowing all patients access, which resulted in an increase in the proportion of atorvastatin dispensed over the following years and a drop in the proportion of simvastatin dispensed. Demographic data were taken from the first statin dispensing record in PHARMS.

Hospital discharge records dating back to 1988 were obtained from the National Minimum Dataset (NMDS) [27]. Each record contains details of the principal and any additional diagnoses, as well as any procedures performed during the hospital stay. Diagnoses are coded to successive editions of the International Statistical Classification of Disease and Related Health Problems, Australian Modification (ICD9-AM and ICD10-AM during the study period) and procedures are coded to the Australian Classification of Health Interventions (ACHI) [28].

The scope of medical practice of statin prescribers, such as General Practice or Internal Medicine, was obtained from the Medical Council of New Zealand (MCNZ) who administer medical practitioners' registrations.

## Identification of the study cohort and primary and secondary prevention groups

The Ministry of Health identified all patients who were dispensed a publicly funded statin between 1 January 2005 and 31 December 2013, and for each patient provided us with anonymised demographic, pharmaceutical, and hospital discharge data. All patient-level data were linked with an encrypted National Health Index, a unique patient identifier used throughout the healthcare system. We excluded patients who received a statin dispensing in 2005 to ensure study members were initiating use for the first time, or following a break of at least one year (Fig 1). Patients who received their first statin dispensing after 31 December 2011 and patients who died within 455 days after their first dispensing were also excluded to ensure there was sufficient follow-up to measure discontinuation. About 15% of patients had at least one statin dispensing record in which the intended duration of the medication supplied (days' supply) was not recorded. For the majority (about 80%) of these patients it was possible to estimate the missing value using information from the patient's other statin dispensing records; the remaining patients were excluded. Patients with only one statin dispensing in the first year of follow-up were also excluded.

We classified patients as belonging to the secondary prevention group if they had a record in the NMDS, prior to the date of their first statin dispensing, of (i) a diagnosis of ischaemic heart disease, ischaemic stroke and/or transient ischaemic attack, and/or (ii) a clinical procedure involving angioplasty of coronary arteries, a stent inserted into the coronary arteries and/or a coronary artery bypass. A full list of diagnosis and procedure codes can be found in S1 and S2 Tables. The remaining patients were classified as belonging to the primary prevention

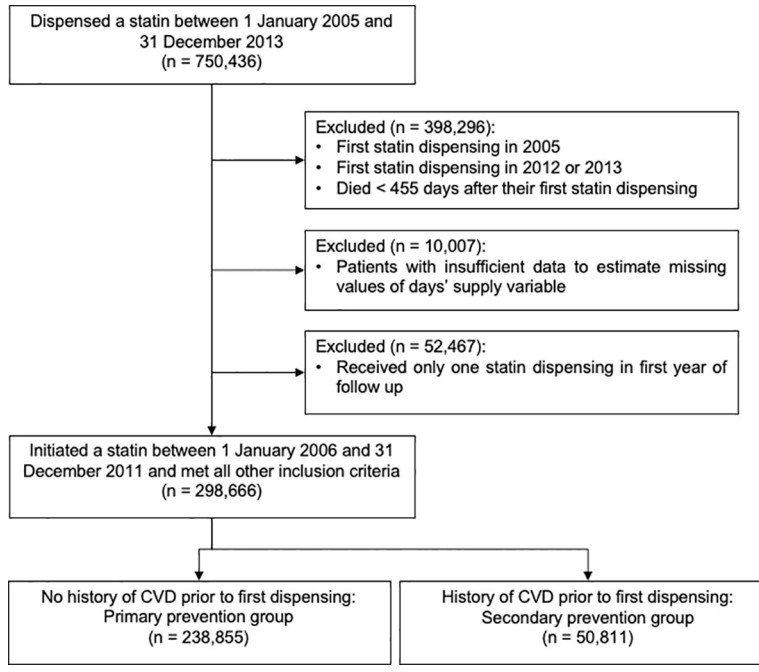

**Fig 1. Identification of the study cohort.**

group. For both groups, the date of the first statin dispensing was taken as the patient's cohort entry date. Patients in the primary prevention group who had a CVD event in the follow-up period were censored.

## Statin exposure

The coverage period for each dispensing was calculated by adding the recorded days' supply to the dispensing date. If a new dispensing occurred during the coverage period of the previous dispensing, we assumed the new supply began at the end of the coverage period. The daily dose of the first statin dispensed was calculated by multiplying the strength of the tablet dispensed by the number of tablets prescribed per day. Because statins are prescribed at different strengths depending on the type of statin [8], the daily dose was divided by the 2018 Defined Daily Dose (DDD) for the type of first statin dispensed (30 mg for simvastatin and pravastatin, 20 mg for atorvastatin) to calculate a DDD ratio. The DDD is an assumed average maintenance dose per day for a medication used for its main indication in adults [29].

## Other key variables

Other key variables included age at first statin dispensing, gender, prioritised ethnicity (using the Statistics New Zealand prioritised order of Māori; Pacific Peoples; Asian; Middle Eastern, Latin American, and African [MELAA]; Other; and European [30]) and NZDep06 [31] (an area-based index of socioeconomic deprivation). The scope of practice for each medical practitioner at the time of the first statin dispensing was identified using data from the MCNZ (S3 Table). If a medical practitioner had multiple scopes of practice, the following hierarchy was used to assign a prioritised scope: Urgent Care (vocational or provisional vocational), Internal Medicine (vocational or provisional vocational), General Practice (vocational or provisional vocational), Other vocation or provisional vocation, General Scope, and Provisional General Scope.

A modified Charlson Comorbidity Index (CCI) [32, 33] that did not include myocardial infarction, peripheral vascular disease, cerebrovascular disease, peptic ulcer disease, or diabetes without chronic complications was calculated at cohort entry using hospital discharge records from the past 5 years.

## Statistical methods

Descriptive statistics were calculated for both the primary and secondary prevention groups. Adherence and discontinuation were examined in the first year of follow-up. The Medication Possession Ratio (MPR) was calculated by dividing the total days of statin coverage in the first year by 365 days. In cases where the supply from a dispensing extended beyond the 365-day mark, the excess supply was truncated. A patient with an MPR $\geq 0.8$ was classified as adherent [34]. Discontinuation was defined as a gap of $> 90$ days between the end of a coverage period and the next dispensing date or the end of the study period, whichever occurred first. Ninety days was chosen as this is the maximum number of days which can be supplied on a prescription in New Zealand. Patients were classified as having discontinued in the first year of follow-up if the start date of the first discontinuation period occurred $< 365$ days after cohort entry.

A lapse in coverage was defined as a gap $\leq 90$ days between the end of the first dispensed supply and the start of the second. Any patients with a period of discontinuation before a lapse of coverage in the first year of follow-up were not eligible for inclusion in the analyses that sought to ascertain whether a lapse of coverage was associated with subsequent discontinuation.

Multivariable logistic regression was used to estimate odds ratios (ORs) and confidence intervals (95% CI) in analyses comparing adherence and discontinuation in the primary versus secondary prevention groups. ORs were adjusted for gender, age at first dispensing, prioritised ethnicity, NZDep06, modified CCI score, year of first statin dispensing, scope of practice of first statin prescriber, first statin dispensed, DDD, and days' supply of first statin dispensing.

### Ethical approval

The study was approved by the University of Otago Human Ethics Committee (Health) (reference: HD15/031). The study was based on routinely collected anonymised data and no patient consent was required.

### Results

Overall, 289,666 new statin users between 2006 and 2011 with a minimum of two dispensings in the first year were identified, of whom 50,811 (17.5%) were hospitalised for CVD prior to cohort entry (the secondary prevention group) (Fig 1). The remaining 238,855 (82.5%) patients had no history of CVD and were assumed to have been dispensed a statin for primary prevention of CVD.

The baseline characteristics of the primary and secondary prevention groups are presented in Table 1. Compared with the primary prevention group, the secondary prevention group was older, had higher proportions of males and patients identifying as European, and higher CCI scores. However, the groups did not differ by socioeconomic status. The secondary prevention group had a higher initial dose of statins and a shorter days' supply but there was no difference between groups in the type of statin first dispensed. The prescriber of the first statin dispensed was more likely to have a General Practice vocational registration in the primary prevention group, compared to the secondary prevention group where doctors with a General Scope registration were the top prescribers. The number of patients initiating a statin for secondary prevention dropped between 2006 and 2011, while the number dispensed statins for primary prevention fluctuated during the same period.

Table 2 presents the mean MPR, adherence, and discontinuation in the first year of follow-up in the two groups. The mean MPR and the proportion adherent in the first year of follow-up were higher in the secondary versus the primary prevention group, while the proportion who discontinued was lower. Compared with the primary prevention group, the secondary prevention group was 1.56 (95% CI 1.52–1.60, p < 0.0001) times as likely to be adherent to statins and 0.67 (95% CI 0.65–0.69, p < 0.0001) times as likely to discontinue statin treatment.

Table 3 presents the mean MPR and adherence for primary and secondary prevention groups according to patient characteristics and details of the first statin dispensing. Adherence was slightly higher for females than males in the primary prevention group but lower in the secondary prevention group. Adherence in both groups increased with age and increasing comorbidity score. Compared to people of European ethnicity, adherence was lower among Māori, Pacific Peoples, and Asians in both the primary and secondary prevention groups. Patients from the lowest socioeconomic quintile had slightly lower adherence than those in the highest quintile in the primary prevention group but not in the secondary prevention group. No difference in adherence between type of statin was seen in the primary prevention group, but patients in the secondary prevention group who were first prescribed atorvastatin were slightly less adherent than those prescribed simvastatin. Adherence was lower in patients in the primary prevention group with a first dispensing lasting less than 30 days. In the primary prevention group, patients whose first statin was prescribed by a doctor with an Internal Medicine vocational registration had the highest adherence, while in the secondary prevention

**Table 1. Baseline characteristics of primary and secondary prevention groups as measured at first statin dispensing.** Values are numbers (percentages) unless stated otherwise.

| Characteristic | Prevention group | |
|---|---|---|
| | Primary (n = 238,855) | Secondary (n = 50,811) |
| **Gender** | | |
| Male | 128,088 (53.6) | 30,705 (60.4) |
| Female | 110,740 (46.4) | 20,105 (39.6) |
| Unspecified | 27 (0.0) | 1 (0.0) |
| **Age at first dispensing (years)** | | |
| < 35 | 6,165 (2.6) | 251 (0.5) |
| 35–44 | 24,009 (10.1) | 2,275 (4.5) |
| 45–54 | 58,069 (24.3) | 7,470 (14.7) |
| 55–64 | 75,062 (31.4) | 12,228 (24.1) |
| 65–74 | 51,705 (21.6) | 12,810 (25.2) |
| ≥ 75 | 23,845 (10.0) | 15, 777 (31.1) |
| Median (IQR) | 59 (50–67) | 67 (57–77) |
| **Ethnicity, prioritised**[*] | | |
| European | 143,107 (59.9) | 38,646 (76.1) |
| Māori | 20,367 (8.5) | 4,762 (9.4) |
| Pacific Peoples | 14,461 (6.1) | 1,968 (3.9) |
| Asian | 19,840 (8.3) | 1,823 (3.6) |
| MELAA[†] | 16,291 (6.8) | 1,611 (3.2) |
| Other | 99 (0.0) | 14 (0.0) |
| Unknown | 24,690 (10.3) | 1,987 (3.9) |
| **NZDep06 quintile** | | |
| 1 (least deprived) | 33,359 (14.0) | 6,407 (12.6) |
| 2 | 35,360 (14.8) | 6,931 (13.6) |
| 3 | 44,643 (18.7) | 9,624 (18.9) |
| 4 | 50,701 (21.2) | 11,680 (23.0) |
| 5 (most deprived) | 59,215 (24.8) | 12,719 (25.0) |
| Unknown | 15,577 (6.5) | 3,450 (6.8) |
| **Modified CCI score**[‡] | | |
| 0 | 219,976 (92.1) | 33,775 (66.5) |
| 1 | 7,294 (3.1) | 3,932 (7.7) |
| 2 | 8,284 (3.5) | 8,434 (16.6) |
| 3 | 1,542 (0.6) | 2,276 (4.5) |
| ≥ 4 | 1,759 (0.7) | 2,394 (4.7) |
| **Year of first statin dispensing** | | |
| 2006 | 40,021 (16.8) | 11,960 (23.5) |
| 2007 | 36,578 (15.3) | 9,623 (18.9) |
| 2008 | 42,284 (17.7) | 8,358 (16.4) |
| 2009 | 45,969 (19.2) | 7,828 (15.4) |
| 2010 | 39,262 (16.4) | 6,714 (13.2) |
| 2011 | 34,741 (14.5) | 6,328 (12.5) |
| **Scope of practice of first statin prescriber** | | |
| Vocational: General Practice | 141,186 (59.1) | 13,045 (25.7) |
| Provisional General Scope | 13,502 (5.7) | 15,187 (29.9) |
| General Scope | 67,130 (28.1) | 17,864 (35.2) |
| Vocational: Internal Medicine[§] | 9,142 (3.8) | 2,058 (4.1) |

(*Continued*)

**Table 1.** (Continued)

| Characteristic | Prevention group | |
|---|---|---|
| | Primary (n = 238,855) | Secondary (n = 50,811) |
| Vocational: Urgent Care | 3,516 (1.5) | 277 (0.5) |
| Vocational: Other | 1,359 (0.6) | 318 (0.6) |
| Unknown | 2,826 (1.2) | 2,022 (4.0) |
| Non-doctor[‖] | 194 (0.1) | 40 (0.1) |
| **First statin dispensed and daily dose (mg/day)** [**] | | |
| Simvastatin[††] | 211,491 (88.5) | 45,006 (88.6) |
| < 20 | 34,414 (16.3) | 2,567 (5.7) |
| 20–39 | 124,781 (59.0) | 15,207 (33.8) |
| 40–59 | 50,523 (23.9) | 26,346 (58.5) |
| 60–79 | 318 (0.2) | 133 (0.3) |
| ≥ 80 | 1,455 (0.7) | 753 (1.7) |
| DDD Ratio (Mean, SD) | 0.79 (0.37) | 1.08 (0.41) |
| Atorvastatin | 27,323 (11.4) | 5,795 (11.4) |
| < 20 | 11,336 (41.5) | 1,120 (19.3) |
| 20–39 | 11,795 (43.2) | 1,559 (26.9) |
| 40–59 | 3,640 (13.3) | 2,179 (37.6) |
| 60–79 | 44 (0.2) | 42 (0.7) |
| ≥ 80 | 508 (1.9) | 895 (15.4) |
| DDD Ratio (Mean, SD) | 0.98 (0.64) | 1.76 (1.13) |
| Pravastatin | 41 (0.0) | 10 (0.0) |
| < 20 | 15 (36.6) | 6 (60.0) |
| 20–39 | 19 (46.3) | 3 (30.0) |
| 40–59 | 5 (12.2) | 1 (10.0) |
| 60–79 | 2 (4.9) | 0 (0.0) |
| ≥ 80 | 15 (36.6) | 6 (60.0) |
| DDD Ratio (Mean, SD) | 0.73 (0.54) | 0.53 (0.32) |
| **Days' supply of first statin dispensing** | | |
| ≤ 30 | 49,387 (20.7) | 24,877 (49.0) |
| 31–60 | 5,999 (2.5) | 1,239 (2.4) |
| 61–90 | 18,3208 (76.7) | 24,651 (48.5) |
| ≥ 91 | 261 (0.1) | 44 (0.1) |

DDD, Defined Daily Dose; SD, Standard deviation.

[*] In cases where patients reported multiple ethnicities, prioritised ethnicity was determined using the Statistics New Zealand prioritised order of Māori, Pacific Peoples, Asian, MELAA, Other, European [30].

[†] Middle Eastern, Latin American, or African.

[‡] CCI weights modified [33] removing myocardial infarction, peripheral vascular disease, cerebrovascular disease, peptic ulcer disease, and diabetes without chronic complications.

[§] Includes Internal Medicine, Cardiology, Clinical Immunology, Clinical Pharmacology, Endocrinology, Gastroenterology, Geriatric Medicine, Haematology, Infectious Diseases, Medical Oncology, Nephrology, Neurology, Nuclear Medicine, Palliative Medicine, Respiratory Medicine and Rheumatology.

[‖] Dentists and Registered nurses working in primary care can prescribe statins.

[**] Denominator for overall proportions by statin type is total number of people in the relevant prevention group, whereas denominator for dosage proportion is the number of people in the relevant prevention group who were taking the statin.

[††] Includes ezetimibe with simvastatin.

**Table 2. Mean MPR, adherence, and discontinuation in primary and secondary prevention groups in first year of follow-up.**

| Prevention group | Mean MPR | Adherent (MPR ≥ 0.8) | | | Discontinued | | |
|---|---|---|---|---|---|---|---|
| | | Adherent (%) | Unadjusted OR (95% CI) | Adjusted OR* (95% CI) | Discontinued (%) | Unadjusted OR (95% CI) | Adjusted OR* (95% CI) |
| Primary | 0.81 | 62.8 | 1.00 | 1.00 | 29.8 | 1.00 | 1.00 |
| Secondary | 0.87 | 76.1 | 1.89 (1.84–1.93) | 1.55 (1.51–1.59) | 19.7 | 0.58 (0.57–0.59) | 0.67 (0.65–0.69) |

MPR, Medication Possession Ratio. OR, Odds Ratio. CI, Confidence interval.

*Adjusted for gender, age at first dispensing, prioritised ethnicity, NZDep06, modified Charlson comorbidity score, year of first statin dispensing, scope of practice of first statin prescriber, first statin dispensed, DDD ratio, and days' supply of first statin dispensing.

group adherence was higher when the statin was prescribed by a doctor with a Provisional General Scope (most practitioners in this category are junior doctors working in hospitals) or Unknown registration.

The analysis of discontinuation of statin therapy is presented in Table 4. Patterns of discontinuation by demographic characteristics and details of the first statin dispensing were similar to those found for adherence in Table 3.

The results of the analyses which explored the impact of a lapse in coverage between the first and second dispensings on subsequent discontinuation are shown in Table 5. The analyses were based on 266,420 patients (after the exclusion of patients with an episode of discontinuation before a lapse of coverage in the first year of follow-up), with 217,679 (81.7%) and 48,741 (18.3%) receiving a statin for primary and secondary prevention, respectively. In total, 103,703 (47.6%) patients in the primary prevention group had a lapse in coverage (median 12 days, interquartile range 4–31 days) between the first and second dispensings, and were 1.51 (CI 95% 1.48–1.54, $p < 0.0001$) times as likely to discontinue than patients with no lapse. In the secondary prevention group, 13,732 (28.2%) patients had a lapse in coverage (median 8 days, interquartile range 3–25 days) and were 1.60 (CI 95% 1.52–1.69, $p < 0.0001$) times as likely to discontinue than patients with no lapse.

## Discussion

In this nationwide study of new users of statins, we found that patients who had been prescribed a statin for secondary prevention of CVD had higher adherence and lower discontinuation than patients prescribed statins for primary prevention. Within the primary prevention group, adherence and discontinuation levels differed by age, comorbidity level, ethnicity and initial prescriber scope of practice. In the secondary prevention group, adherence and discontinuation also varied by age, comorbidity level and ethnicity, as well as initial prescriber scope of practice. However, the type of statin initially dispensed was also associated with both adherence and discontinuation in the secondary prevention group. These patterns of associations remained unchanged when an MPR ≥ 0.90 was used as a threshold for adherence in additional sensitivity analyses (not shown). Of particular note, we found that a lapse in coverage between the first and second dispensings was a predictor of subsequent discontinuation of statin therapy.

Our finding that 76% of patients in the secondary prevention group had an MPR ≥ 0.8 in the first year of follow-up is consistent with results from previous New Zealand investigations [23–25]. A national study of patients aged 35–84 years who were discharged from hospital in 2007 following an admission for angina or acute coronary syndrome (ACS) found that 59% of patients had a statin dispensing ratio (SDR) ≥ 0.8 in the first year of follow-up [25], while a

**Table 3. Mean MPR and adherence in primary and secondary prevention groups in first year of follow-up, by patient characteristics.**

| Characteristic | Primary prevention group | | | | Secondary prevention group | | | |
|---|---|---|---|---|---|---|---|---|
| | Mean | Adherent (MPR ≥ 0.80) | | | Mean | Adherent (MPR ≥ 0.80) | | |
| | MPR | Adherent (%) | Unadjusted OR (95% CI) | Adjusted OR* (95% CI) | MPR | Adherent (%) | Unadjusted OR (95% CI) | Adjusted OR* (95% CI) |
| **Gender** | | | | | | | | |
| Male | 0.80 | 61.1 | 1.00 | 1.00 | 0.87 | 76.8 | 1.00 | 1.00 |
| Female | 0.82 | 64.9 | 1.18 (1.16–1.20) | 1.10 (1.08–1.12) | 0.86 | 75.1 | 0.91 (0.88–0.95) | 0.90 (0.86–0.94) |
| Unspecified | 0.86 | 74.1 | 1.82 (0.81–4.64) | 2.04 (0.88–5.27) | - | - | - | - |
| **Age at first dispensing (years)** | | | | | | | | |
| < 35 | 0.70 | 39.1 | 0.34 (0.32–0.36) | 0.40 (0.38–0.42) | 0.77 | 58.6 | 0.47 (0.37–0.61) | 0.51 (0.40–0.66) |
| 35–44 | 0.74 | 47.8 | 0.49 (0.47–0.50) | 0.55 (0.53–0.57) | 0.82 | 65.9 | 0.65 (0.59–0.71) | 0.67 (0.61–0.74) |
| 45–54 | 0.78 | 55.9 | 0.68 (0.66–0.69) | 0.72 (0.70–0.73) | 0.84 | 71.3 | 0.83 (0.78–0.89) | 0.85 (0.80–0.91) |
| 55–64 | 0.82 | 65.3 | 1.00 | 1.00 | 0.86 | 74.9 | 1.00 | 1.00 |
| 65–74 | 0.85 | 71.5 | 1.33 (1.30–1.37) | 1.30 (1.26–1.34) | 0.88 | 78.1 | 1.20 (1.13–1.27) | 1.19 (1.12–1.26) |
| ≥ 75 | 0.86 | 74.5 | 1.55 (1.50–1.60) | 1.45 (1.40–1.50) | 0.88 | 79.5 | 1.30 (1.23–1.38) | 1.25 (1.18–1.33) |
| **Ethnicity, prioritised** | | | | | | | | |
| European | 0.83 | 67.4 | 1.00 | 1.00 | 0.87 | 78.1 | 1.00 | 1.00 |
| Māori | 0.76 | 51.8 | 0.52 (0.51–0.54) | 0.63 (0.61–0.65) | 0.82 | 66.4 | 0.55 (0.52–0.59) | 0.63 (0.59–0.68) |
| Pacific Peoples | 0.72 | 43.4 | 0.37 (0.36–0.38) | 0.47 (0.45–0.49) | 0.80 | 62.6 | 0.47 (0.43–0.52) | 0.53 (0.48–0.59) |
| Asian | 0.78 | 54.7 | 0.58 (0.57–0.60) | 0.69 (0.67–0.71) | 0.84 | 71.1 | 0.69 (0.62–0.77) | 0.73 (0.66–0.81) |
| MELAA | 0.82 | 63.0 | 0.83 (0.80–0.85) | 0.89 (0.86–0.93) | 0.87 | 77.2 | 0.95 (0.84–1.07) | 0.99 (0.88–1.11) |
| Other | 0.85 | 71.7 | 1.23 (0.80–1.93) | 1.28 (0.83–2.03) | 0.89 | 85.7 | 1.68 (0.46–10.81) | 1.98 (0.54–12.80) |
| Unknown | 0.82 | 63.6 | 0.85 (0.82–0.87) | 0.89 (0.87–0.92) | 0.88 | 78.1 | 1.00 (0.90–1.12) | 1.00 (0.89–1.11) |
| **NZDep06 quintile** | | | | | | | | |
| 1 (least deprived) | 0.83 | 65.6 | 1.00 | 1.00 | 0.87 | 77.6 | 1.00 | 1.00 |
| 2 | 0.82 | 64.9 | 0.97 (0.94–1.00) | 1.01 (0.97–1.04) | 0.87 | 77.7 | 1.01 (0.93–1.09) | 1.02 (0.94–1.10) |
| 3 | 0.82 | 65.0 | 0.98 (0.95–1.00) | 1.01 (0.98–1.04) | 0.87 | 77.0 | 0.97 (0.9–1.04) | 0.99 (0.92–1.07) |
| 4 | 0.82 | 63.9 | 0.93 (0.91–0.96) | 1.00 (0.97–1.03) | 0.87 | 76.7 | 0.95 (0.89–1.02) | 1.00 (0.93–1.08) |
| 5 (most deprived) | 0.79 | 57.8 | 0.72 (0.70–0.74) | 0.91 (0.88–0.94) | 0.85 | 73.3 | 0.79 (0.74–0.85) | 0.94 (0.87–1.01) |
| Unknown | 0.81 | 61.8 | 0.85 (0.82–0.88) | 0.95 (0.92–0.99) | 0.86 | 76.1 | 0.92 (0.84–1.02) | 1.01 (0.91–1.11) |
| **Modified Charlson comorbidity score at first statin dispensing** | | | | | | | | |
| 0 | 0.81 | 62.4 | 1.00 | 1.00 | 0.86 | 75.4 | 1.00 | 1.00 |
| 1 | 0.81 | 64.4 | 1.09 (1.04–1.15) | 1.18 (1.13–1.25) | 0.86 | 74.9 | 0.97 (0.90–1.05) | 1.00 (0.93–1.08) |
| 2 | 0.84 | 70.8 | 1.46 (1.40–1.54) | 1.38 (1.32–1.46) | 0.88 | 78.4 | 1.19 (1.12–1.26) | 1.12 (1.11–1.25) |
| 3 | 0.84 | 70.5 | 1.44 (1.29–1.61) | 1.49 (1.33–1.67) | 0.87 | 78.0 | 1.15 (1.04–1.28) | 1.18 (1.06–1.31) |
| ≥ 4 | 0.85 | 72.4 | 1.58 (1.42–1.76) | 1.49 (1.34–1.66) | 0.87 | 78.1 | 1.16 (1.05–1.28) | 1.17 (1.05–1.29) |
| **Year of first statin dispensing** | | | | | | | | |
| 2006 | 0.81 | 62.8 | 1.00 | 1.00 | 0.86 | 74.6 | 1.00 | 1.00 |
| 2007 | 0.81 | 62.7 | 1.00 (0.97–1.03) | 1.00 (0.97–1.03) | 0.86 | 76.3 | 1.09 (1.03–1.16) | 1.08 (1.01–1.15) |
| 2008 | 0.81 | 62.9 | 1.00 (0.98–1.03) | 1.02 (0.99–1.05) | 0.86 | 75.9 | 1.07 (1.00–1.14) | 1.05 (0.99–1.12) |
| 2009 | 0.82 | 63.3 | 1.02 (0.99–1.05) | 1.04 (1.01–1.07) | 0.87 | 76.3 | 1.09 (1.02–1.17) | 1.08 (1.01–1.15) |
| 2010 | 0.81 | 62.6 | 0.99 (0.96–1.02) | 1.02 (0.99–1.05) | 0.87 | 77.7 | 1.19 (1.11–1.27) | 1.17 (1.09–1.26) |
| 2011 | 0.81 | 62.7 | 0.99 (0.96–1.02) | 1.04 (1.00–1.07) | 0.87 | 77.0 | 1.14 (1.06–1.22) | 1.13 (1.04–1.22) |
| **Scope of practice of first statin prescriber** | | | | | | | | |
| Vocational: General Practice | 0.82 | 64.0 | 1.00 | 1.00 | 0.86 | 73.5 | 1.00 | 1.00 |

*(Continued)*

**Table 3.** (Continued)

| Characteristic | Primary prevention group | | | | Secondary prevention group | | | |
| --- | --- | --- | --- | --- | --- | --- | --- | --- |
| | Mean | Adherent (MPR $\geq$ 0.80) | | | Mean | Adherent (MPR $\geq$ 0.80) | | |
| | MPR | Adherent (%) | Unadjusted OR (95% CI) | Adjusted OR* (95% CI) | MPR | Adherent (%) | Unadjusted OR (95% CI) | Adjusted OR* (95% CI) |
| Provisional General Scope | 0.80 | 63.5 | 0.98 (0.94–1.02) | 0.95 (0.91–0.98) | 0.87 | 78.1 | 1.29 (1.22–1.36) | 1.21 (1.14–1.28) |
| General Scope | 0.80 | 59.8 | 0.84 (0.82–0.86) | 0.90 (0.89–0.92) | 0.87 | 76.3 | 1.16 (1.10–1.22) | 1.13 (1.07–1.20) |
| Vocational: Internal Medicine | 0.85 | 70.9 | 1.37 (1.31–1.44) | 1.24 (1.18–1.30) | 0.87 | 75.9 | 1.14 (1.02–1.27) | 1.10 (0.98–1.22) |
| Vocational: Urgent Care | 0.76 | 51.8 | 0.61 (0.57–0.65) | 0.81 (0.75–0.86) | 0.83 | 69.7 | 0.83 (0.64–1.08) | 0.93 (0.72–1.22) |
| Vocational: Other | 0.79 | 61.2 | 0.89 (0.80–0.99) | 1.04 (0.93–1.17) | 0.85 | 70.8 | 0.87 (0.69–1.12) | 0.82 (0.64–1.06) |
| Unknown | 0.81 | 64.5 | 1.02 (0.95–1.11) | 1.01 (0.93–1.09) | 0.87 | 78.1 | 1.28 (1.15–1.44) | 1.23 (1.10–1.38) |
| Non–doctor | 0.80 | 59.3 | 0.82 (0.62–1.10) | 0.89 (0.67–1.20) | 0.88 | 80.0 | 1.44 (0.70–3.36) | 1.34 (0.64–3.14) |
| **First statin dispensed** | | | | | | | | |
| Simvastatin | 0.81 | 62.8 | 1.00 | 1.00 | 0.87 | 76.2 | 1.00 | 1.00 |
| Atorvastatin | 0.81 | 63.1 | 1.01 (0.99–1.04) | 0.98 (0.95–1.01) | 0.87 | 75.9 | 0.99 (0.93–1.05) | 0.86 (0.79–0.93) |
| Pravastatin | 0.80 | 65.9 | 1.14 (0.61–2.24) | 1.37 (0.72–2.73) | 0.67 | 60.0 | 0.47 (0.13–1.84) | 0.54 (0.15–2.13) |
| **DDD ratio of first statin dispensed** | | | 0.94 (0.93–0.96) | 0.98 (0.96–1.00) | | | 1.16 (1.12–1.21) | 1.23 (1.18–1.29) |
| **Days' supply of first statin dispensing** | | | | | | | | |
| $\leq$ 30 | 0.76 | 59.4 | 0.83 (0.81–0.85) | 0.80 (0.78–0.82) | 0.86 | 76.7 | 1.05 (1.01–1.09) | 0.94 (0.90–0.98) |
| 31–60 | 0.79 | 62.1 | 0.93 (0.88–0.98) | 0.85 (0.80–0.89) | 0.84 | 70.0 | 0.74 (0.66–0.84) | 0.70 (0.62–0.80) |
| 61–90 | 0.83 | 63.8 | 1.00 | 1.00 | 0.88 | 75.8 | 1.00 | 1.00 |
| $\geq$ 91 | 0.92 | 79.7 | 2.23 (1.66–3.04) | 2.18 (1.62–2.98) | 0.92 | 81.8 | 1.43 (0.70–3.32) | 1.40 (0.68–3.25) |

MPR, Medication Possession Ratio. OR, Odds Ratio. CI, Confidence interval. DDD, Defined Daily Dose.

*Adjusted for gender, age at first dispensing, prioritised ethnicity, NZDep06, modified Charlson comorbidity score, year of first statin dispensing, scope of practice of first statin prescriber, first statin dispensed, DDD ratio, and days' supply of first statin dispensing.

related national study confined to patients discharged following an ACS admission in the same year found that 69% of patients had an MPR $\geq$ 0.8 in the first year of follow-up [23]. In the third study by the same research group, 75% of patients discharged from two hospitals following ACS in the years 2007–2011 had an MPR $\geq$ 0.8 in the first year of follow-up [24]. These investigations differed somewhat from our study in that they were confined to patients with acute coronary events, and SDR and MPR values were calculated for all patients, including those who were not dispensed a statin in the first year of follow-up. To the best of our knowledge, this study is the first to provide national-level information about adherence among New Zealand patients dispensed statins for primary prevention, so no direct comparison with previous work is possible.

In contrast to our nationwide study, research on statin adherence and discontinuation in primary and secondary prevention internationally has focussed on specific sub-populations or samples from the general population. Nonetheless, the differences we observed between primary and secondary prevention groups is consistent with other research internationally. For instance, higher levels of discontinuation and lower adherence in primary versus secondary prevention have been reported among diverse patient groups including middle-aged members of a health insurance plan in Quebec, Canada [35], elderly patients in Ontario, Canada [36], patients over 45 in Finland [37], patients enrolled in managed care organisations in the United States [38, 39], members of a health maintenance organisation in Israel [40], and patients

**Table 4.** Discontinuation in primary and secondary prevention groups in first year of follow–up, by patient characteristics.

| Characteristic | Primary prevention group | | | Secondary prevention group | | |
|---|---|---|---|---|---|---|
| | Discontinued (%) | Unadjusted OR (95% CI) | Adjusted OR* (95% CI) | Discontinued (%) | Unadjusted OR (95% CI) | Adjusted OR* (95% CI) |
| **Gender** | | | | | | |
| Male | 30.6 | 1.00 | 1.00 | 18.4 | 1.00 | 1.00 |
| Female | 28.8 | 0.91 (0.90–0.93) | 0.97 (0.95–0.99) | 21.8 | 1.24 (1.19–1.30) | 1.23 (1.17–1.28) |
| Unspecified | 18.5 | 0.51 (0.17–1.25) | 0.47 (0.16–1.17) | – | – | – |
| **Age at first statin dispensing (years)** | | | | | | |
| < 35 | 51.7 | 2.86 (2.72–3.02) | 2.48 (2.36–2.62) | 37.8 | 2.46 (1.90–3.18) | 2.32 (1.78–3.00) |
| 35–44 | 41.5 | 1.90 (1.84–1.96) | 1.71 (1.66–1.76) | 27.0 | 1.49 (1.35–1.66) | 1.47 (1.32–1.63) |
| 45–54 | 34.7 | 1.42 (1.38–1.45) | 1.35 (1.31–1.38) | 22.6 | 1.18 (1.10–1.27) | 1.17 (1.09–1.25) |
| 55–64 | 27.2 | 1.00 | 1.00 | 19.8 | 1.00 | 1.00 |
| 65–74 | 23.4 | 0.81 (0.79–0.84) | 0.83 (0.81–0.85) | 18.0 | 0.89 (0.83–0.95) | 0.88 (0.82–0.94) |
| ≥ 75 | 22.2 | 0.76 (0.74–0.79) | 0.80 (0.77–0.83) | 18.4 | 0.91 (0.86–0.97) | 0.90 (0.84–0.96) |
| **Ethnicity, prioritised*** | | | | | | |
| European | 26.3 | 1.00 | 1.00 | 18.4 | 1.00 | 1.00 |
| Māori | 38.6 | 1.76 (1.71–1.81) | 1.48 (1.43–1.53) | 27.0 | 1.64 (1.53–1.76) | 1.45 (1.34–1.56) |
| Pacific Peoples | 44.9 | 2.28 (2.20–2.36) | 1.84 (1.78–1.91) | 29.9 | 1.89 (1.71–2.09) | 1.71 (1.54–1.90) |
| Asian | 36.9 | 1.63 (1.58–1.69) | 1.42 (1.37–1.46) | 23.8 | 1.38 (1.24–1.54) | 1.33 (1.19–1.49) |
| MELAA[+] | 29.0 | 1.14 (1.10–1.18) | 1.09 (1.05–1.13) | 16.9 | 0.90 (0.79–1.03) | 0.90 (0.78–1.02) |
| Other | 23.2 | 0.85 (0.52–1.33) | 0.80 (0.49–1.25) | 7.1 | 0.34 (0.02–1.71) | 0.29 (0.02–1.47) |
| Unknown | 28.4 | 1.11 (1.08–1.15) | 1.08 (1.05–1.11) | 17.4 | 0.93 (0.83–1.05) | 0.96 (0.85–1.08) |
| **NZ Dep06 quintile** | | | | | | |
| 1 (least deprived) | 14.0 | 1.00 | 1.00 | 12.6 | 1.00 | 1.00 |
| 2 | 14.8 | 1.06 (1.02–1.09) | 1.02 (0.99–1.06) | 13.6 | 1.01 (0.93–1.10) | 1.00 (0.92–1.09) |
| 3 | 18.7 | 1.07 (1.04–1.10) | 1.03 (1.00–1.07) | 18.9 | 1.05 (0.97–1.14) | 1.02 (0.94–1.11) |
| 4 | 21.2 | 1.11 (1.07–1.14) | 1.03 (0.99–1.06) | 23.0 | 1.05 (0.97–1.14) | 0.99 (0.92–1.08) |
| 5 (most deprived) | 24.8 | 1.38 (1.34–1.42) | 1.11 (1.08–1.15) | 25.0 | 1.28 (1.19–1.38) | 1.11 (1.02–1.20) |
| Unknown | 6.5 | 1.17 (1.12–1.22) | 1.06 (1.01–1.10) | 6.8 | 1.08 (0.97–1.20) | 1.01 (0.91–1.13) |
| **Modified Charlson comorbidity score at first statin dispensing** | | | | | | |
| 0 | 30.1 | 1.00 | 1.00 | 20.0 | 1.00 | 1.00 |
| 1 | 29.4 | 0.97 (0.92–1.02) | 0.87 (0.84–0.93) | 21.5 | 1.09 (1.01–1.19) | 1.04 (0.96–1.13) |
| 2 | 24.4 | 0.75 (0.71–0.79) | 0.77 (0.73–0.81) | 18.4 | 0.90 (0.85–0.96) | 0.88 (0.83–0.94) |
| 3 | 24.2 | 0.74 (0.66–0.83) | 0.70 (0.62–0.79) | 19.3 | 0.96 (0.86–1.06) | 0.90 (0.81–1.01) |
| ≥ 4 | 24.0 | 0.74 (0.66–0.82) | 0.76 (0.68–0.85) | 18.6 | 0.91 (0.82–1.02) | 0.87 (0.78–0.97) |
| **Year of first statin dispensing** | | | | | | |
| 2006 | 29.5 | 1.00 | 1.00 | 20.2 | 1.00 | 1.00 |
| 2007 | 29.6 | 1.01 (0.98–1.04) | 1.00 (0.97–1.03) | 19.7 | 0.97 (0.91–1.04) | 0.98 (0.91–1.05) |
| 2008 | 29.8 | 1.01 (0.98–1.05) | 1.00 (0.97–1.03) | 20.0 | 0.99 (0.92–1.06) | 1.00 (0.93–1.07) |
| 2009 | 29.3 | 0.99 (0.96–1.02) | 0.98 (0.95–1.01) | 19.7 | 0.97 (0.91–1.05) | 0.97 (0.90–1.05) |
| 2010 | 30.0 | 1.02 (0.99–1.05) | 1.00 (0.97–1.03) | 19.0 | 0.93 (0.86–1.00) | 0.93 (0.86–1.01) |
| 2011 | 30.6 | 1.05 (1.02–1.09) | 1.02 (0.98–1.05) | 19.4 | 0.95 (0.88–1.03) | 0.98 (0.90–1.07) |
| **Scope of practice of first statin prescriber** | | | | | | |
| Vocational: General Practice | 28.4 | 1.00 | 1.00 | 21.1 | 1.00 | 1.00 |
| Provisional General Scope | 31.3 | 1.15 (1.11–1.19) | 1.18 (1.13–1.23) | 18.5 | 0.85 (0.80–0.90) | 0.91 (0.86–0.97) |
| General Scope | 32.6 | 1.22 (1.20–1.25) | 1.14 (1.12–1.17) | 19.7 | 0.91 (0.86–0.96) | 0.95 (0.90–1.01) |

(*Continued*)

**Table 4.** (Continued)

| Characteristic | Primary prevention group | | | Secondary prevention group | | |
|---|---|---|---|---|---|---|
| | Discontinued (%) | Unadjusted OR (95% CI) | Adjusted OR* (95% CI) | Discontinued (%) | Unadjusted OR (95% CI) | Adjusted OR* (95% CI) |
| Vocational: Internal Medicine | 24.5 | 0.82 (0.78–0.86) | 0.91 (0.87–0.96) | 20.1 | 0.94 (0.84–1.05) | 0.99 (0.88–1.12) |
| Vocational: Urgent Care | 38.8 | 1.60 (1.49–1.71) | 1.24 (1.16–1.33) | 25.6 | 1.29 (0.97–1.68) | 1.15 (0.87–1.51) |
| Vocational: Other | 33.7 | 1.28 (1.15–1.44) | 1.12 (1.00–1.26) | 23.0 | 1.11 (0.85–1.44) | 1.21 (0.92–1.57) |
| Unknown | 29.1 | 1.04 (0.95–1.12) | 1.05 (0.97–1.14) | 18.7 | 0.86 (0.76–0.97) | 0.92 (0.81–1.04) |
| Non–doctor | 35.1 | 1.36 (1.01–1.82) | 1.25 (0.92–1.68) | 22.5 | 1.08 (0.49–2.18) | 1.16 (0.52–2.36) |
| **First statin dispensed** | | | | | | |
| Simvastatin | 29.8 | 1.00 | 1.00 | 19.8 | 1.00 | 1.00 |
| Atorvastatin | 29.9 | 1.01 (0.98–1.03) | 1.03 (0.99–1.06) | 19.2 | 0.96 (0.90–1.03) | 1.14 (1.05–1.24) |
| Pravastatin | 19.5 | 0.57 (0.25–1.18) | 0.46 (0.19–0.96) | 50.0 | 4.05 (1.13–14.57) | 3.23 (0.89–11.74) |
| **DDD ratio of first statin dispensed** | – | 1.02 (1.00–1.04) | 0.98 (0.96–1.00) | – | 0.78 (0.75–0.81) | 0.75 (0.70–0.78) |
| **Days' supply of first statin dispensing** | | | | | | |
| ≤ 30 | 33.2 | 1.22 (1.19–1.25) | 1.23 (1.20–1.26) | 19.6 | 0.99 (0.94–1.03) | 1.04 (0.99–1.09) |
| 31–60 | 28.3 | 0.97 (0.91–1.03) | 1.04 (0.98–1.10) | 19.9 | 1.00 (0.87–1.15) | 1.03 (0.89–1.19) |
| 61–90 | 28.9 | 1.00 | 1.00 | 19.9 | 1.00 | 1.00 |
| ≥ 91 | 22.6 | 0.72 (0.53–0.95) | 0.74 (0.55–0.98) | 15.9 | 0.76 (0.31–1.61) | 0.75 (0.31–1.60) |

OR, Odds Ratio. CI, Confidence interval. DDD, Defined Daily Dose.

*Adjusted for gender, age at first dispensing, prioritised ethnicity, NZDep06, modified Charlson comorbidity score, year of first statin dispensing, scope of practice of first statin prescriber, first statin dispensed, DDD ratio, and days' supply of first statin dispensing.

attending practices contributing to a general practice research database in the United Kingdom [41].

Our finding that statin adherence differed by age, gender, ethnicity, and comorbidities is consistent with findings from elsewhere [16, 34]. With the exception of gender, we found that these factors influenced statin adherence similarly in both primary and secondary prevention groups. Our observation that, after adjusting for other factors, atorvastatin was associated with slightly lower adherence than simvastatin in the secondary prevention group is consistent with the findings of a study in the United States [42]. Conversely, a study in the United Kingdom based on a secondary prevention group found no differences in adherence to atorvastatin and simvastatin [43], and Finnish research reported adherence to be higher for atorvastatin than simvastatin in both a primary prevention [44] and mixed-prevention cohort [37]. In contrast

**Table 5. Impact of a lapse of coverage between the first and second statin dispensing on odds of subsequent discontinuation of statin.**

| | Primary prevention group | | | Secondary prevention group | | |
|---|---|---|---|---|---|---|
| | Discontinued (%) | Unadjusted OR (95% CI) | Adjusted OR* (95% CI) | Discontinued (%) | Unadjusted OR (95% CI) | Adjusted OR* (95% CI) |
| Lapse of coverage between first and second dispensing | 27.0 | 1.56 (1.53–1.59) | 1.51 (1.48–1.54) | 21.2 | 1.59 (1.51–1.67) | 1.60 (1.52–1.69) |
| No lapse of coverage | 19.2 | 1.00 | 1.00 | 14.4 | 1.00 | 1.00 |

OR, Odds Ratio. CI, Confidence interval.

*Adjusted for gender, age at first dispensing, prioritised ethnicity, NZDep06, modified Charlson comorbidity score, year of first statin dispensing, scope of practice of first statin prescriber, first statin dispensed, DDD ratio, and days' supply of first statin dispensing.

to some overseas studies [44–46], we did not find an association between socioeconomic status and statin adherence except for patients living in the most deprived areas. This may partly reflect the variation in how socioeconomic status is measured across countries and studies.

Our study had several strengths and limitations. One strength is that our study uses a cohort of all patients in New Zealand who initiated statin therapy during the study period, thereby minimising selection bias and improving statistical power. For a community pharmacist to be reimbursed for a dispensing they must submit a claim that is recorded in PHARMS. This means the statin dispensing data for funded statins are likely to be virtually complete for the whole of New Zealand during the study period. We believe rosuvastatin, a non-funded statin, is rarely prescribed and its non-inclusion will not impact our results. The use of the NMDS allowed us to identify patients with a history of being admitted to hospital for CVD as all public hospitals in New Zealand are required to report all inpatient and day patient discharges with diagnosis data.

Our study was based on dispensing information and we did not have access to prescription data, therefore patients who were prescribed a statin but never filled their prescription would not have been included in our cohort. However, the lack of prescribing data may not have been an issue for our estimates of adherence in the secondary prevention group as previous New Zealand research has found that most patients who were discharged from hospital following an acute coronary syndrome event filled their statin prescription [24]. The use of dispensing data to calculate adherence is based on an assumption that medication possession equates to consumption, however this limitation applies to nearly all methods of measuring medication adherence. Our classification of patients into primary and secondary prevention groups was based on hospital discharge records. A recent study found 39% of patients with a history of CVD hospitalisation did not have that history recorded in their primary care notes [47]. This missing information might have an impact on the type and strength of statin the medical practitioner prescribes and the type of conversation they have around the reason for taking the statin. Finally, we did not assess the impact of other factors which may influence adherence behaviour, such as depression and alcohol use [16, 48, 49], due to the absence or poor capture of information on these factors in the data available to us.

Our data do not explain why adherence was lower in the primary prevention group than in the secondary prevention group. The motivation to take a statin regularly is likely to be higher in the latter group due to their having already experienced one of the events taking a statin is intended to prevent. The absolute benefits of statin therapy also increase as cardiovascular risk increases while the risk of statin-induced side effects remains largely static regardless of cardiovascular risk, leading to a better benefit-risk profile in secondary prevention patients and greater motivation [38, 40].

There are some important implications for statin prescribers, and the wider healthcare delivery system, from our study. The most novel is that the late filling of a statin prescription is associated with subsequent discontinuation in the first year of therapy. This delay therefore provides an important indicator for clinicians to identify and intervene early in patients at increased risk of statin therapy discontinuation. Our finding of differences in the levels of adherence and risk of discontinuation across population groups based on prescription refill data speak not only to differences in adherence behaviour between these groups, but also to issues with access to medicines for these groups. While much is often discussed about how medication adherence might be improved by interventions targeted at individuals' adherence behaviour, our results are also consistent with other research that has identified barriers in the equitable access to healthcare for Māori and Pacific populations in particular within New Zealand [50–52]. These groups are also at significantly higher risk of death from cardiac events [53], and it is incumbent on the health system to consider not just the behavioural

interventions which can be used to improve adherence in these populations, but the way in which the system itself delivers healthcare [54].

In conclusion, our study showed adherence to statin medication is lower in primary prevention than secondary prevention, is associated with demographic factors, and discontinuation of medication can be predicted by late filling of the second prescription. Using this information, strategies can be developed to identify patients at increased risk of poor adherence early in their statin therapy and guide further research into how healthcare can be better delivered to increase patient adherence and reduce the risk of CVD.

## Supporting information

**S1 Table. ICD codes for cardiovascular disease diagnoses.**
(DOCX)

**S2 Table. ACHI codes for cardiovascular disease procedures.**
(DOCX)

**S3 Table. Medical Council of New Zealand (MCNZ) definitions.**
(DOCX)

## Acknowledgments

The authors thank Analytical Services at the New Zealand Ministry of Health for providing the necessary data and the Medical Council of New Zealand for providing the medical scope of practice.

## Author Contributions

**Conceptualization:** Finn Sigglekow, Simon Horsburgh, Lianne Parkin.

**Data curation:** Finn Sigglekow, Simon Horsburgh.

**Formal analysis:** Finn Sigglekow.

**Funding acquisition:** Simon Horsburgh, Lianne Parkin.

**Methodology:** Finn Sigglekow, Simon Horsburgh, Lianne Parkin.

**Project administration:** Simon Horsburgh, Lianne Parkin.

**Resources:** Simon Horsburgh, Lianne Parkin.

**Software:** Finn Sigglekow.

**Supervision:** Simon Horsburgh, Lianne Parkin.

**Writing – original draft:** Finn Sigglekow.

**Writing – review & editing:** Simon Horsburgh, Lianne Parkin.

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
