## [Decision Letter · Decision Letter 0]

26 Aug 2020

PONE-D-20-14591

Statin adherence is lower in primary than secondary prevention: a national follow-up study of new users

PLOS ONE

Dear Dr. Horsburgh,

Thank you for submitting your manuscript to PLOS ONE. After careful consideration, we feel that it has merit but does not fully meet PLOS ONE’s publication criteria as it currently stands. Therefore, we invite you to submit a revised version of the manuscript that addresses the points raised during the review process.

Please revise your comments based on the reviewers suggestions. Note reviewer 1 suggests commenting on the differences in associations between socioeconomic status and adherence that may differ to other countries without universal health care. There are more substantive suggestions from reviewer 2 that should be addressed.

We look forward to receiving your revised manuscript.

Kind regards,

Seana Gall

Academic Editor

PLOS ONE

Journal Requirements:

2. Our internal editors have evaluated your manuscript and determined that it is within the scope of our 'Primary and Secondary Prevention of Cardiovascular Disease' Call for Papers. This collection of papers is headed by a team of Guest Editors for PLOS ONE and will encompass a diverse range of research articles. Additional information can be found on our announcement page: (https://collections.plos.org/s/prevention-cardiovascular). If you would like your manuscript to be considered for this collection, please let us know in your cover letter and we will ensure that your paper is treated as if you were responding to this call. If you would prefer to remove your manuscript from collection consideration, please specify this in the cover letter.

3. We noted in your submission details that a portion of your manuscript may have been presented or published elsewhere.

"Parts of this research were presented at: (i) the 243rd Otago Medical School Research Society Summer Student Speaker Meeting, Dunedin, New Zealand on 8 May 2018, and (ii) the 4th Pharmacoepidemiology Research Network Symposium, Dunedin, New Zealand on 21 November 2018 as verbal presentations, and an abstract published in the accompanying proceedings.  The manuscript itself, or the research in its totality, has not been published or is pending publication."

Please clarify whether this publication was peer-reviewed and formally published. If this work was previously peer-reviewed and published, in the cover letter please provide the reason that this work does not constitute dual publication and should be included in the current manuscript.

Reviewers' comments:

Reviewer's Responses to Questions

**Comments to the Author**

1. Is the manuscript technically sound, and do the data support the conclusions?

Reviewer #1: Yes

Reviewer #2: Yes

2. Has the statistical analysis been performed appropriately and rigorously? 

Reviewer #1: Yes

Reviewer #2: Yes

3. Have the authors made all data underlying the findings in their manuscript fully available?

Reviewer #1: Yes

Reviewer #2: Yes

4. Is the manuscript presented in an intelligible fashion and written in standard English?

Reviewer #1: Yes

Reviewer #2: Yes

5. Review Comments to the Author

Reviewer #1: Statin nonadherence is an important problem due to association with increased risk of CVD and death. Study was well done using standard methods. A new user design was used to investigate statin adherence in a very large (n>289,000) 6-year sample of New Zealand community pharmacies. Statins are recommended by NZ guidelines, and paid for by national health insurance. Nonetheless, Consistent with other studies, they found secondary prevention statin uses were more adherent than primary prevention users, and adherence was influenced by several demographic factors (age, sex, race/ethnicity, comorbidity).

A more novel finding was that an early gap in statin coverage increased the risk of statin discontinuation. It should also be noted that 90 day prescriptions were associated with increased adherence, so perhaps this one way to address this barrier. One interesting finding that was not highlighted by authors was the lack of association of socioeconomic status with nonadherence. This has implications for other countries, such as US, which do not have national health insurance.

Reviewer #2: This was a retrospective nationwide study using dispensing data from New Zealand community pharmacies to compare statin adherence and discontinuation in primary and secondary prevention patients. Comments were provided as follows:

Comments by section:

Abstract

• Add ‘and discontinuation’ after ‘We undertook a nationwide study to compare adherence’.

Introduction

• In the first paragraph, add what risk equation is used to estimate 5-year CVD risk in the NZ guidelines.

• In the first paragraph, add data to show the prevalence and incidence of statin use in NZ if there are any.

• In the second paragraph, consider saying “The effectiveness of statin therapy is limited by a patient’s level of adherence (the degree to which a patient conforms to a prescribed course of medication[10]) in patient groups at varying CVD risks. Several studies found a relationship between poor statin adherence, the degree of lipid lowering, and poor CVD outcomes in both primary [ref] and secondary prevention populations [ref].’

• The second paragraph is a bit sketchy. Consider adding a new paragraph illustrating how the non-adherence to and discontinuation of statin treatment influence patient’s CVD outcomes based on the existing evidence base and the negative consequences of statin nonadherence and discontinuation (i.e. increased healthcare costs).

Methods

• Although the MPR = 80% is generally acceptable as a cut-off to define medication adherence, it remains somewhat arbitrary. The authors may consider adding a sensitivity analysis to compare the adherence rate between the primary and secondary prevention groups using different cut-offs of ‘adherence’.

Results

• The adjusted OR and 95% CI for statin adherence (OR 1.55, 95% CI, 1.51-1.59, Table 2) does not match the estimates reported in the abstract and main text (OR 1.56, 95% CI, 1.52-1.60).

• In Table 5, the adjusted OR for secondary prevention group does not match the estimate reported in the main text as well. Please carefully check other data.

• In Tables 3 & 4, for the subgroup analysis by age, consider setting the age <35 group as the reference group so that the results can be presented in a more intuitive way. (I presume that the age was treated as a categorical variable in your logistic models). Ditto day’s supply of first statin dispensing.

• In terms of ethnicity, scope of practice of first statin prescriber, and gender, consider moving the reference group to the top.

• Table 5- the ORs and 95%CIs went to the different lines.

Discussion

• In the first paragraph, do you mean ‘research comparing statin adherence and discontinuation between the primary and secondary prevention populations elsewhere has focused on sub-groups within populations’? Can you specify this as many studies investigated the statin adherence in general populations?

• In the first paragraph, the authors wrote “Within primary and secondary prevention groups, adherence and discontinuation differed by demographic characteristics and details of the first statin dispensed”. Can you specify what demographic characteristics and details of the first statin dispensed you referred to?

• In the second paragraph, add “to our best knowledge” before “Our study is the first to provide national-level information”

• Paragraph 3- The suboptimal statin adherence in primary prevention populations relative to the secondary prevention populations is not something new. The authors may want to add their own thoughts to explain why this happened. For example, the benefits of statin therapy accrue with increased CVD risk while the risk of statin-induced side effects typically distribute equally over patients with varying levels of CVD risk.

• Paragraph 7: could make this sentence more clear by saying “Our data do not explain why adherence was lower in the primary prevention group than in the secondary prevention group, however the motivation to take a statin regularly is likely to be higher in the latter due to their higher risk of a recurrent cardiovascular event and mortality.” Also add a reference at the end. // you may want to delete this sentence if you have addressed another similar comment mentioned above.

• Can you add a couple of sentences somewhere to describe the implication of your study findings to clinical practice or healthcare system? And What value does this manuscript add to existing evidence base?

• Study strengths - I think the major strength of this study is use of nationally representative sample, so that the analyses had sufficient study power and an ability to provide the estimates regarding statin use in the entire NZ population. You may want to add this strength somewhere.

• Study limitations: - The inability to assess the statin adherence and discontinuation by other unmeasured/unobserved patient characteristics (such as a healthy lifestyle, psychological factors) is worth mentioning.

Conclusion

• The authors stated that ‘using this information, strategies can be developed to increase patient adherence and reduce risk of CVD’. This conclusion is a bit sketchy. Consider improving it.

References:

• Fix the reference 33.

6. PLOS authors have the option to publish the peer review history of their article (what does this mean?). If published, this will include your full peer review and any attached files.

Reviewer #1: No

Reviewer #2: No

---

## [Author Response · Author response to Decision Letter 0]

19 Oct 2020

Responses to Reviewers for Manuscript PONE-D-20-14591: Statin adherence is lower in primary than secondary prevention: a national follow-up study of new users.

We would like to thank the reviewers for their thoughtful and constructive comments and suggestions on the manuscript. Please find our responses to these below.

Editorial Queries

We have reviewed the manuscript and ensured that it meets PLOS One’s style requirements.

We have addressed the other editorial queries in the accompanying cover letter as requested.

 

Reviewer #1

1. One interesting finding that was not highlighted by authors was the lack of association of socioeconomic status with nonadherence. This has implications for other countries, such as US, which do not have national health insurance.

The reviewer makes an interesting observation, and the lack of an association between adherence and socioeconomic status has been quite consistent in other research we have conducted in New Zealand (e.g. [1,2]). While New Zealand typically requires very low co-payments from patients for their medications, New Zealanders do have to potentially pay much more for general practitioner visits. We have been involved in some research examining how much people pay for their medicines across a range of health systems but this work is still being written up for publication. One clear learning from that work is that the role of how countries approach the public and private contribution to medication costs has on medication is likely to be complex and beyond the scope of this manuscript to address meaningfully. It is very much an important topic that does need to be further examined.

Another point that may be important to consider when looking at the lack of a gradient between socioeconomic status and adherence in this manuscript compared to studies in other countries is the differing ways of measuring socioeconomic status. We used the standard tool in New Zealand, the New Zealand Deprivation Index (NZDep) [3]. The NZDep is a multidimensional, area-based measure which is a weighted composite of ten factors associated with socioeconomic status. While income, educational level and work status are heavily weighted elements of the NZDep, other elements such as telephone access, household occupancy (e.g. support) and access to a car are also important factors. In contrast, many overseas studies (such as [4–6]) use income, education level and potentially work status separately at an individual level as proxies for socioeconomic status. The differences in the findings between countries may therefore at least partially reflect the varied ways in which socioeconomic status is measured across studies. We have added the following to the Discussion to highlight this point:

In contrast to some overseas studies [44–46], we did not find an association between socioeconomic status and statin adherence except for patients living in the most deprived areas. This may partly reflect the variation in how socioeconomic status is measured across countries and studies.

 

Reviewer #2

Abstract

1. Add ‘and discontinuation’ after ‘We undertook a nationwide study to compare adherence’.

Thank you for this suggestion. We have made the suggested change in the Abstract.

Introduction

2. In the first paragraph, add what risk equation is used to estimate 5-year CVD risk in the NZ guidelines.

We have adopted this suggestion and expanded the relevant sentence in the Introduction to provide this information. We have also added a reference to a recent article which includes information on the risk prediction equations used in the guidelines historically and into the future (one of the authors was the person responsible for those parts of the guidelines). The sentence now reads:

Successive versions of New Zealand primary care guidelines have recommended statins for primary prevention in patients with no history of CVD disease if their 5-year CVD risk is above 15% using risk prediction equations based on the Framingham Heart Study [6], and for secondary prevention following angina, myocardial infarction, ischaemic stroke, or transient ischaemic attack [7–10].

3. In the first paragraph, add data to show the prevalence and incidence of statin use in NZ if there are any.

We have added this information using information on dispensings and the usually resident populations, and added citations for these data sources. The revised sentence in the first paragraph of the Introduction now reads:

Statins have become a mainstay of the pharmacological prevention of CVD in New Zealand; in 2015 514,277 people (14.6% of the New Zealand population aged 18 or over) were dispensed a statin and this number has continued to rise [11,12].

4. In the second paragraph, consider saying “The effectiveness of statin therapy is limited by a patient’s level of adherence (the degree to which a patient conforms to a prescribed course of medication[10]) in patient groups at varying CVD risks. Several studies found a relationship between poor statin adherence, the degree of lipid lowering, and poor CVD outcomes in both primary [ref] and secondary prevention populations [ref].’

Thank you for this suggestion. Please see our response to point 5 below, as we have included the suggested amendment into our response to that point.

5. The second paragraph is a bit sketchy. Consider adding a new paragraph illustrating how the non-adherence to and discontinuation of statin treatment influence patient’s CVD outcomes based on the existing evidence base and the negative consequences of statin nonadherence and discontinuation (i.e. increased healthcare costs).

We have expanded the second paragraph to incorporate the suggestions in point 4 and addressing the concerns raised here. The paragraph now reads:

The effectiveness of statin therapy is limited by a patient’s level of adherence (the degree to which a patient conforms to a prescribed course of medication [13]). Research has demonstrated greater lipid control with higher statin adherence levels [14]. A protective relationship between statin adherence and CVD outcomes has also been found in both primary and secondary prevention populations in a number of observational studies [15–19]. In addition to the health outcome benefits accruing from the effectiveness of statin therapy, high levels of adherence lead to reduced future healthcare costs through decreased hospitalisations [20–22]. Collectively, improving statin adherence leads to better patient and health system outcomes.

Methods

6. Although the MPR = 80% is generally acceptable as a cut-off to define medication adherence, it remains somewhat arbitrary. The authors may consider adding a sensitivity analysis to compare the adherence rate between the primary and secondary prevention groups using different cut-offs of ‘adherence’.

Thank you for suggesting this additional analysis. We re-ran our analyses using a cut-off for adherence of MPR ≥ 0.90. The results of these analyses are summarised in Table 1 at the end of this document. For simplicity and ease of comparing with Table 3 in the manuscript, we have left out the mean MPR (since this would not change) and the unadjusted odds ratio (95% confidence interval) columns. As expected, the percentage of patients who were adherent dropped markedly. However, the adjusted odds ratios for adherence in the secondary versus primary prevention population group remained virtually identical (MPR ≥ 0.90 OR = 1.55 [95% CI: 1.51 – 1.58], MPR ≥ 0.80 OR = 1.55 [95% CI: 1.51 – 1.59]), while the odds ratios for the various patient characteristics also only changed very slightly. Adopting the stricter criterion of an MPR ≥ 0.90 does not appear to affect the pattern of findings.

We have added the following sentence in the first paragraph of the Discussion to indicate that increasing the MPR cut-off did not change the findings:

These patterns of associations remained unchanged when an MPR ≥ 0.90 was used as a threshold for adherence in additional sensitivity analyses (not shown).

Results

7. The adjusted OR and 95% CI for statin adherence (OR 1.55, 95% CI, 1.51-1.59, Table 2) does not match the estimates reported in the abstract and main text (OR 1.56, 95% CI, 1.52-1.60).

Thank you for noticing this. We have double-checked the figures and made sure that the correct ones are reported in the abstract, text and table.

8. In Table 5, the adjusted OR for secondary prevention group does not match the estimate reported in the main text as well. Please carefully check other data.

Thank you again for bringing these errors to our attention. We have corrected the figure quoted in the text and also re-checked figures in the tables. This has resulted in a minor change to some 95% confidence intervals reported in Tables 3 and 5 (where a digit was left off) which has no impact on the results or their interpretation.

9. In Tables 3 & 4, for the subgroup analysis by age, consider setting the age <35 group as the reference group so that the results can be presented in a more intuitive way. (I presume that the age was treated as a categorical variable in your logistic models). Ditto day’s supply of first statin dispensing.

The reference categories were chosen to reflect the usual age at which people are dispensed statins/amount that is usually dispensed. This is a ‘normative’ approach to choosing the reference category. While choosing, for example, the < 35 age band as the reference group might make the presentation of results clearer, all of the coefficients from the regression models would be in comparison to a group which typically is far less likely to receive a statin in the first place. We believe it is more informative for researchers and clinicians alike to be presented with odds ratios representing how a particular age group differs from the norm (e.g. ‘do patients who present at an older age than typical have better/worse adherence?’ vs ‘do patients older than 75 have better/worse adherence than those < 35?’ The age gradient of adherence is still easily seen by the reader through the ‘Mean MPR’ columns. The same point holds for days’ supply of first statin dispensing, where the vast majority of patients are dispensed a 90-day supply of statins.

10. In terms of ethnicity, scope of practice of first statin prescriber, and gender, consider moving the reference group to the top.

We have made this suggested amendment to Tables 3 and 5. We have also amended Table 1 to make the ordering consistent throughout the manuscript.

11. Table 5- the ORs and 95%CIs went to the different lines.

Thank you for noticing this. We have reformatted the page so that Table 5 now fits as it should without spilling columns across lines.

Discussion

12. In the first paragraph, do you mean ‘research comparing statin adherence and discontinuation between the primary and secondary prevention populations elsewhere has focused on sub-groups within populations’? Can you specify this as many studies investigated the statin adherence in general populations?

We think that the reviewer is actually referring to paragraph three. The original wording was ambiguous. What we intended to convey was that there is a dearth of research internationally which compares statin adherence in primary and secondary populations using an entire population. Instead, these studies assess statin adherence in either specific populations (people with diabetes, women, people enrolled in particular insurance schemes) or a sample from the general population. This was notable from Hope et al.’s recent systematic review of statin adherence for primary prevention [7]. We have amended the wording at the beginning of paragraph three to clarify this point:

In contrast to our nationwide study, research on statin adherence and discontinuation in primary and secondary prevention internationally has focussed on specific sub-populations or samples from the general population.

13. In the first paragraph, the authors wrote “Within primary and secondary prevention groups, adherence and discontinuation differed by demographic characteristics and details of the first statin dispensed”. Can you specify what demographic characteristics and details of the first statin dispensed you referred to?

We have expanded the first paragraph of the Discussion to incorporate the reviewer’s suggestion. The paragraph now reads:

In this nationwide study of new users of statins, we found that patients who had been prescribed a statin for secondary prevention of CVD had higher adherence and lower discontinuation than patients prescribed statins for primary prevention. Within the primary prevention group, adherence and discontinuation levels differed by age, comorbidity level, ethnicity and initial prescriber scope of practice. In the secondary prevention group, adherence and discontinuation also varied by age, comorbidity level and ethnicity, as well as initial prescriber scope of practice. However, the type of statin initially dispensed was also associated with both adherence and discontinuation in the secondary prevention group. Of particular note, we found that a lapse in coverage between the first and second dispensings was a predictor of subsequent discontinuation of statin therapy.

14. In the second paragraph, add “to our best knowledge” before “Our study is the first to provide national-level information”

Thank you for this suggestion. We have revised the relevant sentence to read:

To the best of our knowledge, this study is the first to provide national-level information about adherence among New Zealand patients dispensed statins for primary prevention, so no direct comparison with previous work is possible.

15. Paragraph 3- The suboptimal statin adherence in primary prevention populations relative to the secondary prevention populations is not something new. The authors may want to add their own thoughts to explain why this happened. For example, the benefits of statin therapy accrue with increased CVD risk while the risk of statin-induced side effects typically distribute equally over patients with varying levels of CVD risk.

This is an important point, and we have addressed this in our response to point 16 below.

16. Paragraph 7: could make this sentence more clear by saying “Our data do not explain why adherence was lower in the primary prevention group than in the secondary prevention group, however the motivation to take a statin regularly is likely to be higher in the latter due to their higher risk of a recurrent cardiovascular event and mortality.” Also add a reference at the end. // you may want to delete this sentence if you have addressed another similar comment mentioned above.

Thank you for this suggestion. We have added the following paragraph to the Discussion in response to point 15 above and point 16:

Our data do not explain why adherence was lower in the primary prevention group than in the secondary prevention group. The motivation to take a statin regularly is likely to be higher in the latter group due to their having already experienced one of the events taking a statin is intended to prevent. The absolute benefits of statin therapy also increase as cardiovascular risk increases while the risk of statin-induced side effects remains largely static regardless of cardiovascular risk, leading to a better benefit-risk profile in secondary prevention patients and greater motivation [38,40]. 

17. Can you add a couple of sentences somewhere to describe the implication of your study findings to clinical practice or healthcare system? And What value does this manuscript add to existing evidence base?

We feel the reviewer makes an excellent suggestion here, and have added a paragraph into the Discussion to address it. The paragraph reads:

There are some important implications for statin prescribers, and the wider healthcare delivery system, from our study. The most novel is that the late filling of a statin prescription is associated with subsequent discontinuation in the first year of therapy. This delay therefore provides an important indicator for clinicians to identify and intervene early in patients at increased risk of statin therapy discontinuation. Our finding of differences in the levels of adherence and risk of discontinuation across population groups based on prescription refill data speak not only to differences in adherence behaviour between these groups, but also to issues with access to medicines for these groups. While much is often discussed about how medication adherence might be improved by interventions targeted at individuals’ adherence behaviour, our results are also consistent with other research that has identified barriers in the equitable access to healthcare for Māori and Pacific populations in particular within New Zealand [50–52]. These groups are also at significantly higher risk of death from cardiac events [53], and it is incumbent on the health system to consider not just the behavioural interventions which can be used to improve adherence in these populations, but the way in which the system itself delivers healthcare [54].

18. Study strengths - I think the major strength of this study is use of nationally representative sample, so that the analyses had sufficient study power and an ability to provide the estimates regarding statin use in the entire NZ population. You may want to add this strength somewhere.

We appreciate the reviewer suggesting emphasising an important strength of our study. We have added the following sentence to the strengths section of the Discussion:

One strength is that our study uses a cohort of all patients in New Zealand who initiated statin therapy during the study period, thereby minimising selection bias and improving statistical power.

19. Study limitations: - The inability to assess the statin adherence and discontinuation by other unmeasured/unobserved patient characteristics (such as a healthy lifestyle, psychological factors) is worth mentioning.

We agree with the reviewer that it is important to mention this point in our limitations. We have added the following sentence to the limitations section of our Discussion:

Finally, we did not assess the impact of other factors which may influence adherence behaviour, such as depression and alcohol use [16,48,49], due to the absence or poor capture of information on these factors in the data available to us.

20. Conclusion - The authors stated that ‘using this information, strategies can be developed to increase patient adherence and reduce risk of CVD’. This conclusion is a bit sketchy. Consider improving it.

Thank you for highlighting the need to improve this conclusion. We have made it more explicit and amended it to read:

Using this information, strategies can be developed to identify patients at increased risk of poor adherence early in their statin therapy and guide further research into how healthcare can be better delivered to increase patient adherence and reduce the risk of CVD.

References

21. Fix the reference 33.

Thank you for noticing this error. We have fixed the reference in the reference list, and also reviewed the other references to make sure that they conform with the Journal’s style requirements.

 

References

1. Horsburgh S, Norris P, Becket G, Arroll B, Crampton P, Cumming J, et al. Allopurinol use in a New Zealand population: prevalence and adherence. Rheumatol Int. 2014;34: 963–970. doi:10.1007/s00296-013-2935-5

2. Horsburgh S, Barson D, Zeng J, Sharples K, Parkin L. Adherence to metformin monotherapy in people with type 2 diabetes mellitus in New Zealand. Diabetes Res Clin Pract. 2019;158: 107902. doi:10.1016/j.diabres.2019.107902

3. Salmond C, Crampton P, Atkinson J. NZDep2006 Index of Deprivation user’s manual. Wellington: University of Otago; 2007. 

4. Aarnio E, Martikainen J, Winn AN, Huupponen R, Vahtera J, Korhonen MJ. Socioeconomic inequalities in statin adherence under universal coverage: does sex matter? Circ Cardiovasc Qual Outcomes. 2016;9: 704–713. doi:10.1161/CIRCOUTCOMES.116.002728

5. Erickson SR, Bravo M, Tootoo J. Geosocial Factors Associated With Adherence to Statin Medications. Ann Pharmacother. 2020;54: 1194–1202. doi:10.1177/1060028020934879

6. Xie Z, St. Clair P, Goldman DP, Joyce G. Racial and ethnic disparities in medication adherence among privately insured patients in the United States. Ruiz JM, editor. PLoS One. 2019;14: e0212117. doi:10.1371/journal.pone.0212117

7. Hope HF, Binkley GM, Fenton S, Kitas GD, Verstappen SMM, Symmons DPM. Systematic review of the predictors of statin adherence for the primary prevention of cardiovascular disease. Zeeb H, editor. PLoS One. 2019;14: e0201196. doi:10.1371/journal.pone.0201196

 

Tables

Table 1: Adherence in primary and secondary prevention groups in first year of follow-up using MPR cut-off of 0.90, by patient characteristics

Characteristic Primary prevention group Secondary prevention group

 Adherent (MPR > 0.90) Adherent (MPR > 0.90)

 Adherent (%) Adjusted OR*

(95% CI) Adherent (%) Adjusted OR*

(95% CI)

Gender 

 Male 49.7 1.00 67.3 1.00

 Female 54.7 1.14 (1.13 – 1.16) 65.9 0.92 (0.88 – 0.96)

 Unspecified 66.7 2.30 (1.03 – 5.46) - -

Age at first dispensing (years) 

 < 35 29.4 0.42 (0.39 – 0.44) 46.2 0.51 (0.39 – 0.65)

 35 – 44 35.7 0.54 (0.52 – 0.55) 53.3 0.65 (0.59 – 0.71)

 45 – 54 44.0 0.71 (0.70 – 0.73) 60.1 0.84 (0.79 – 0.89)

 55 – 64 54.1 1.00 65.0 1.00

 65 – 74 61.7 1.33 (1.30 – 1.36) 69.6 1.22 (1.15 – 1.28)

 ≥ 75 66.1 1.55 (1.50 – 1.60) 71.2 1.28 (1.21 – 1.35)

Ethnicity, prioritised 

 European 56.8 1.00 69.2 1.00

 Māori 40.0 0.61 (0.59 – 0.63) 54.0 0.61 (0.57 – 0.65)

 Pacific Peoples 31.9 0.45 (0.43 – 0.47) 50.7 0.52 (0.48 – 0.57)

 Asian 43.4 0.69 (0.67 – 0.71) 61.1 0.75 (0.68 – 0.83)

 MELAA 52.1 0.90 (0.87 – 0.93) 67.0 0.95 (0.85 – 1.06)

 Other 59.6 1.16 (0.77 – 1.75) 78.6 1.90 (0.59 – 8.43)

 Unknown 52.5 0.89 (0.87 – 0.92) 69.8 1.03 (0.93 – 1.14)

NZDep06 quintile 

 1 (least deprived) 54.5 1.00 68.5 1.00

 2 53.5 1.00 (0.97 – 1.03) 69.0 1.03 (0.96 – 1.11)

 3 54.4 1.03 (1.00 – 1.06) 68.0 1.00 (0.93 – 1.07)

 4 53.4 1.03 (1.00 – 1.06) 67.3 1.00 (0.94 – 1.07)

 5 (most deprived) 47.1 0.96 (0.92 – 0.99) 63.1 0.94 (0.88 – 1.00)

 Unknown 50.5 0.96 (0.92 – 0.99) 66.7 1.01 (0.92 – 1.10)

Modified Charlson comorbidity score at first statin dispensing

 0 51.5 1.00 65.9 1.00

 1 53.2 1.15 (1.10 – 1.21) 65.6 1.01 (0.94 – 1.09)

 2 61.2 1.40 (1.34 – 1.47) 69.7 1.18 (1.12 – 1.25)

 3 61.2 1.53 (1.38 – 1.71) 67.9 1.12 (1.02 – 1.23)

 ≥ 4 63.3 1.53 (1.38 – 1.69) 68.9 1.15 (1.05 – 1.27)

Year of first statin dispensing

 2006 51.7 1.00 65.0 1.00

 2007 52.1 1.02 (0.99 – 1.05) 66.8 1.07 (1.01 – 1.13)

 2008 52.0 1.03 (1.00 – 1.06) 66.1 1.04 (0.98 – 1.10)

 2009 52.5 1.05 (1.02 – 1.08) 67.6 1.11 (1.04 – 1.18)

 2010 51.7 1.03 (1.00 – 1.06) 68.6 1.16 (1.09 – 1.24)

 2011 52.0 1.07 (1.03 – 1.10) 67.7 1.13 (1.05 – 1.22)

Scope of practice of first statin prescriber

 Vocational: General 

 Practice 53.1 1.00 64.2 1.00

 Provisional General 

 Scope 52.4 0.94 (0.90 – 0.97) 68.9 1.19 (1.12 – 1.25)

 General Scope 49.0 0.91 (0.89 – 0.93) 66.7 1.12 (1.06 – 1.17)

 Vocational: Internal 

 Medicine 60.9 1.25 (1.20 – 1.31) 66.5 1.07 (0.97 – 1.18)

 Vocational: Urgent 

 Care 40.2 0.80 (0.74 – 0.85) 57.8 0.86 (0.67 – 1.10)

 Vocational: Other 50.6 1.06 (0.95 – 1.19) 64.5 0.97 (0.77 – 1.23)

 Unknown 53.7 1.01 (0.93 – 1.09) 68.3 1.18 (1.07 – 1.31)

 Non–doctor 46.9 0.84 (0.63 – 1.13) 75.0 1.60 (0.80 – 3.48)

First statin dispensed

 Simvastatin 52.1 1.00 66.4 1.00

 Atorvastatin 43.9 0.96 (0.94 – 0.99) 50.0 0.88 (0.82 – 0.94)

 Pravastatin 52.0 0.86 (0.46 – 1.65) 66.8 0.56 (0.15 – 2.04)

DDD ratio of first statin dispensed 0.96 (0.94 – 0.98) 1.18 (1.14 – 1.22)

Days’ supply of first statin dispensing 

 ≤ 30 48.1 0.77 (0.76 – 0.79) 66.9 0.90 (0.86 – 0.94)

 31 – 60 49.3 0.77 (0.73 – 0.81) 59.8 0.70 (0.62 – 0.79)

 61 – 90 53.1 1.00 66.9 1.00

 ≥ 91 76.2 2.81 (2.11 – 3.78) 75.0 1.43 (0.74 – 2.99)

MPR, Medication Possession Ratio. OR, Odds Ratio. CI, Confidence interval. DDD, Defined Daily Dose. 

*Adjusted for gender, age at first dispensing, prioritised ethnicity, NZDep06, modified Charlson comorbidity score, year of first statin dispensing, scope of practice of first statin prescriber, first statin dispensed, DDD ratio, and days’ supply of first statin dispensing.

---

## [Decision Letter · Decision Letter 1]

3 Nov 2020

Statin adherence is lower in primary than secondary prevention: a national follow-up study of new users

PONE-D-20-14591R1

Dear Dr. Horsburgh,

We’re pleased to inform you that your manuscript has been judged scientifically suitable for publication and will be formally accepted for publication once it meets all outstanding technical requirements.

Kind regards,

Seana Gall

Academic Editor

PLOS ONE

Additional Editor Comments (optional):

Thank you for your careful consideration and responses to the reviewers comments.

Reviewers' comments:

Reviewer's Responses to Questions

**Comments to the Author**

1. If the authors have adequately addressed your comments raised in a previous round of review and you feel that this manuscript is now acceptable for publication, you may indicate that here to bypass the “Comments to the Author” section, enter your conflict of interest statement in the “Confidential to Editor” section, and submit your "Accept" recommendation.

Reviewer #1: All comments have been addressed

Reviewer #2: All comments have been addressed

2. Is the manuscript technically sound, and do the data support the conclusions?

Reviewer #1: Yes

Reviewer #2: Yes

3. Has the statistical analysis been performed appropriately and rigorously? 

Reviewer #1: Yes

Reviewer #2: Yes

4. Have the authors made all data underlying the findings in their manuscript fully available?

Reviewer #1: Yes

Reviewer #2: Yes

5. Is the manuscript presented in an intelligible fashion and written in standard English?

Reviewer #1: Yes

Reviewer #2: Yes

6. Review Comments to the Author

Reviewer #1: Well done. My comments were adequately addressed. No further revisions needed from my perspective...

Reviewer #2: This paper reads good to me now. I appreciate that the authors have carefully addressed all of my comments.

7. PLOS authors have the option to publish the peer review history of their article (what does this mean?). If published, this will include your full peer review and any attached files.

Reviewer #1: No

Reviewer #2: No

---

## [Editor Report · Acceptance letter]

9 Nov 2020

PONE-D-20-14591R1 

Statin adherence is lower in primary than secondary prevention: a national follow-up study of new users 

Dear Dr. Horsburgh:

I'm pleased to inform you that your manuscript has been deemed suitable for publication in PLOS ONE. Congratulations! Your manuscript is now with our production department. 

Kind regards, 

on behalf of

Dr. Seana Gall 

Academic Editor

PLOS ONE